# Associations of SNPs of the *ADIPOQ* Gene with Serum Adiponectin Levels, Unstable Angina, and Coronary Artery Disease

**DOI:** 10.3390/biom9100537

**Published:** 2019-09-26

**Authors:** Stepan Smetnev, Marina Klimushina, Vladimir Kutsenko, Anna Kiseleva, Nadezhda Gumanova, Alexander Kots, Olga Skirko, Alexandra Ershova, Elena Yarovaya, Victoria Metelskaya, Alexey Meshkov, Oksana Drapkina

**Affiliations:** 1National Medical Research Center for Preventive Medicine, Ministry of Healthcare of the Russian Federation, 101990 Moscow, Russia; stefancom@mail.ru (S.S.); vlakutsenko@ya.ru (V.K.); sanyutabe@gmail.com (A.K.); ngumanova@gnicpm.ru (N.G.); oskirko@gnicpm.ru (O.S.); alersh@mail.ru (A.E.); yarovaya@mech.math.msu.su (E.Y.); vametelskaya@gmail.com (V.M.); meshkov@lipidclinic.ru (A.M.); odrapkina@gnicpm.ru (O.D.); 2Department of Clinical Pharmacology, Sechenov First Moscow State Medical University, 119992 Moscow, Russia; aykots@gmail.com; 3Department of Molecular and Cellular Genetics, Pirogov Russian National Research Medical University, 117997 Moscow, Russia

**Keywords:** adiponectin, *ADIPOQ*, coronary artery disease, single-nucleotide polymorphisms, genetic markers

## Abstract

Adiponectin is encoded by the *ADIPOQ* gene and participates in the pathogenesis of cardiovascular and metabolic diseases. The goal of the study was to assess associations of rs17300539, rs266729, rs182052, rs2241766, and rs17366743 single nucleotide polymorphisms (SNPs) of the *ADIPOQ* gene with concentrations of serum adiponectin and with coronary atherosclerosis and type 2 diabetes mellitus in 447 patients (316 men and 131 women) subjected to coronary angiography. SNPs of the *ADIPOQ* gene of the study participants were genotyped using real-time PCR. Multivariate linear regression adjusted for covariates revealed significant association between rs182052 SNP and serum adiponectin concentration (β= –0.11; 95% confidence interval (95%CI): –0.19, –0.03; *p* = 0.016). Regression analysis revealed an increase in prevalence of unstable angina (OR (odds ratio) = 2.55; 95%CI 1.4–4.82; *p* = 0.018) and coronary artery disease (OR = 1.55; 95%CI 1.15–2.09; *p* = 0.021) per copy of the rs182052 A allele. Prevalence of type 2 diabetes mellitus was higher in subjects with the rs182052 A allele (OR = 2.29; 95%CI 1.29-4.21; *p* = 0.024). Regression analysis of rs266729 showed that prevalence of unstable angina was increased (OR = 3.59; 95%CI 1.17–10.01; *p* = 0.045) in the subjects with the GG genotype and prevalence of coronary artery disease (CAD) was significantly increased (OR = 1.48; 95%CI 1.09–2.03; *p* = 0.045) per copy of the G allele. Haplotype analysis revealed that the subjects with the GCATT haplotype have lower adiponectin levels (β= –0.15; *p* = 0.042) and higher prevalence of unstable angina (OR = 3.597; *p* = 0.007) compared with reference haplotype carriers. Thus, the results indicate that minor A allele of rs182052 of the *ADIPOQ* gene is significantly associated with a decrease in serum adiponectin levels, and two SNPs (rs182052 and rs266729) of the *ADIPOQ* gene are significantly associated with cardiovascular and metabolic diseases.

## 1. Introduction

Adipocytes produce various mediators involved in pathogenesis of cardiovascular diseases. Adiponectin is a 28 kDa protein [1] that plays a crucial role in vascular physiology, controls the metabolism of glucose and fatty acids, modulates insulin sensitivity, suppresses proliferation of vascular smooth muscle cells, and has a general anti-inflammatory effect, thus regulating the important components of atherogenesis [2,3,4].

Adiponectin is encoded by the *ADIPOQ* gene. Activity of the *ADIPOQ* gene is one of the key factors contributing to production of adiponectin. Various single nucleotide polymorphisms (SNPs) of the *ADIPOQ* gene are associated with the development of type 2 diabetes mellitus (T2D) [5], metabolic syndrome [6], and cardiovascular diseases [4,7]. However, associations described in various studies in different populations are variable apparently due to differences in ethnicity of the participating subjects, study design, cohort size, environmental effects, etc. [7,8,9]. Only a few studies have investigated the SNPs of the *ADIPOQ* gene in Russia [10,11]. Thus, associations of the SNPs of the *ADIPOQ* gene with various pathological conditions and factors are poorly understood and require additional investigation.

We have selected five specific SNPs that were expected to influence the circulating adiponectin levels according to the OMIM database (omim.org), provided that their anticipated mean allele frequency (MAF) is more than 1%, in the European population. Associations of the selected SNPs with adiponectin levels have been previously detected in multiple studies [5,12,13,14,15,16,17].

In general, the effects of specific SNPs are relatively small and significant associations are difficult to demonstrate. Moreover, certain SNPs can be linked to other SNPs forming a linkage group. Hence, reliability of SNP haplotype associations with pathological factors can be substantially improved by inclusion of linkage disequilibrium (LD) to identify the most important associations and unfavorable haplotypes.

The goal of the present study was to determine associations of five *ADIPOQ* SNPs (rs17300539, rs266729, rs182052, rs2241766, and rs17366743) with serum levels of adiponectin, coronary artery disease (CAD), and T2D in a cohort of Russian patients who underwent coronary angiography.

## 2. Materials and Methods 

### 2.1. Subjects

At the planning stage, design of the study implied the use of available blood samples from a previously described cohort (N = 500) that included patients treated at the National Research Center for Preventive Medicine, Ministry of Healthcare of Russian Federation (Moscow, Russia) in 2011–2012. Details and characteristics have been published by us previously [18]. A total of 447 patients from this cohort were randomly selected for SNP assessment based on the selection of the SNPs, quality of the blood sample, and specific expected mean allele frequencies to ensure sufficient power of the study. 

The patients were suspected to have coronary artery disease and were subjected to transfemoral coronary angiography by the method of Judkins (GE Innova 4100IQ system, General Electric, Milwaukee, WI, USA) with subsequent transluminal balloon coronary angioplasty and stenting if required. Typical indications for angiography included positive exercise test, positive stress echocardiography, clear symptoms of advanced angina pectoris if there were contraindications for a stress test, and symptoms of unstable angina. Extent of stenosis was quantified with Advantage Workstation software version 4.4 (General Electric, Waukesha, WI, USA). Gensini score was used to estimate the extent of the lesions of coronary arteries [19].

Exclusion criteria included previous myocardial infarction or stroke less than 6 months before the study, any acute inflammatory disease, chronic kidney failure with glomerular filtration rate below 60 mL/min/1.73 m^2^, diabetes mellitus type I, left ventricular ejection fraction below 40%, oncological diseases, hematological diseases with abnormal platelet count, blood coagulation disorders, immune diseases, pregnancy, and lactation.

Coronary artery disease was diagnosed in patients with unstable angina, prior myocardial infarction (over 6 months ago), prior revascularization, and positive stress test in conjunction with the Gensini score ≥ 3. Unstable angina was diagnosed based on the anamnesis, dynamics of angina progression, or signs of ischemia on electrocardiogram [20]. 

T2D was diagnosed according to the WHO guidelines in patients with fasting glucose levels ≥ 7 mmol/L or glycated hemoglobin ≥ 6.5 [21].

The study was compliant with the good clinical practice standards and the principles of the Helsinki Declaration. All participants signed an informed consent prior to enrollment. The study protocol was approved by the Ethics Committee of National Medical Research Center for Preventive Medicine.

### 2.2. Sample Collection and Measurements

The blood was withdrawn from the cubital vein after 12 to 14 h of fasting. Serum was prepared by centrifugation at 1000× *g* for 20 min at 4 °C, aliquoted and stored at −26 °C. Total cholesterol and triglycerides were assayed using a Konelab 20i automatic analyzer (Thermo Electron Corp., Vantaa, Finland) using ELISA kits from Human (Wiesbaden, Germany). High density lipoprotein (HDL) cholesterol was assayed after precipitation of low and very low density lipoproteins. Level of low density lipoprotein (LDL) cholesterol was calculated according to the Friedewald equation in samples with serum triglycerides below 4.5 mmol/L [22]. Blood glucose and high sensitivity C-reactive protein (hsCRP) were tested with kits from DiaSys (Holzheim, Germany) using a Sapphire 400 automatic turbidimetry analyzer (Tokyo, Japan). Insulin was measured by chemiluminescence using an Architect i2000 sr analyzer (Abbot Diagnostics, Lake Bluff, IL, USA). Homeostatic model assessment insulin resistance index (HOMA-IR) was calculated according to Equation (1): [glucose, mmol/L] × [insulin, µIU/mL]/22.5.(1)

Concentration of adiponectin was determined by an ELISA kit (BioVendor, Brno, Czech Republic). 

Genomic DNA was extracted from the whole blood samples using a QIAamp_®_ DNA blood mini kit (QIAGEN, Hilden, Germany) and stored at −20 °C until analysis. DNA concentration was measured by a Nano photometer (Implen, München, Germany). 

### 2.3. SNP Selection

Five SNPs of the *ADIPOQ* gene were selected based on the Exac database (http://exac.broadinstitute.org/) and previous publications [4,5,7,12,13,14,15,16,17] including rs17300539, rs182052, rs266729, rs2241766, and rs17366743, with minor allele frequency (MAF) > 1%. These polymorphisms are distributed throughout the gene and their associations with adiponectin levels or CAD and related phenotypes were demonstrated previously [4,5,7,23]. 

### 2.4. Genotyping

Sequences of the primers and probes are listed in Table 1. Genotypes of five SNPs of the *ADIPOQ* gene (rs17300539, rs182052, rs266729, rs2241766, and rs17366743) were determined using an ABI-7500 fast real-time PCR system (Applied Biosystems, Thermo Fisher Scientific, Foster City, CA, USA) with TaqMan probes from Syntol (Moscow, Russia). Thermal cycling was performed as follows: 63 °C for 1 min and 95 °C for 3 min, followed by 40 cycles at 95 °C for 15 s and 63 °C for 40 s. After PCR amplification, allelic discrimination was performed using an ABI-7500 instrument. Ten percent of the samples were genotyped in duplicates. Real-time PCR data were validated in selected samples (at least three for each genotype) by Sanger sequencing of the PCR products (Genom Medicobiological Center, Moscow, Russia). The results are shown in Figure 1. Hardy–Weinberg equilibrium (HWE) was verified for genotype frequencies by the χ^2^ test [24]. All SNPs were found to be in HWE in the cohort with an exception of rs266729 (*p* = 0.001).

### 2.5. Statistical Analysis 

Statistical data analysis was performed using R version 3.5.1.

Continuous variables with Groeneveld–Meeden (GM) skewness less than 0.2 [25] were approximated by normal distribution and thus are presented as the mean ± SD. Continuous variables with GM skewness over 0.2 are presented as the median and interquartile range. Dichotomous categorical variables are presented as frequency and 95% confidence interval (CI). The concentrations of adiponectin, C-reactive protein, and triglycerides (TG) had the GM skewness more than 0.2, while their logarithms had the GM skewness less than 0.2. Thus, logarithms of these variables were used for calculations.

Associations between logarithm of adiponectin concentration and SNPs were estimated by multivariate linear regression models in different genetic models adjusted for the following covariates: sex, age, LDL and HDL cholesterol, logarithm of TG and hsCRP concentrations, presence of T2D and hypertension, and use of warfarin and statins. A fraction of the data was incomplete (0.2% of hsCRP levels, 0.2% of warfarin use, 3.6% of HDL levels, and less than 1.2% of other data of the lipid profile); these missing values were replaced with the median values [26]. To assess the associations with the genotypes, additive, dominant, and recessive genetic models were used. In the additive model, all SNP genotypes were evaluated separately, including MM, Mm, and mm, where M corresponds to the major allele and m corresponds to the minor allele. In the dominant and recessive models, the heterozygote genotypes were combined with the major or minor genotypes, respectively. In analysis of SNP associations with various parameters, results of multivariate parametric methods were validated with univariate nonparametric methods. *p* < 0.05 was used to indicate statistical significance. The associations with categorical variables representing diseases were calculated as odds ratio (OR) and their significance was estimated by Fisher’s exact test.

Heavily-tailed parametric distributions may be unstable. Therefore, we decided to additionally validate the models using a nonparametric test. Moreover, adjustment for covariates may interfere with assessment of the SNP associations. Specific justification is similar to that outlined previously in a relevant study [27]. 

The results of the regression analysis were validated by the Mann–Whitney test for dominant and recessive models and adjusted for multiple comparisons by the Westfall and Young method [28] using the multtest module for R [29]. The maximum variance inflation factor for the covariates list was 1.4, which indicates no significant linear dependence between the covariates [30]. Associations between the prevalence of various diseases and SNPs were estimated by multivariate logistic regression with adjustment for the previous list of covariates. The results were validated by the exact Fisher test and adjusted for multiple comparisons by the Holm method.

Linkage disequilibrium was visualized using the Haploview software version 4.2 [31]. The haplotypes were constructed using the Plink software [32]. Multivariate linear and logistic regressions with adjustment for previous list of covariates were used to estimate associations of the haplotypes with adiponectin levels and CAD under assumption of additive haplotypes effect. A haplotype with the highest frequency was selected as the reference.

## 3. Results

### 3.1. Subject Characteristics and Genotyping

The study included 447 thoroughly examined subjects; mean age 61 ± 9 years. Baseline clinical assessment, biochemical parameters, diagnosis, and medications are shown in Table 2. A total of 64% (N = 286) of the participants were diagnosed with coronary artery disease. The cohort included patients with unstable angina (4.9%; N = 22), T2D patients (17.2%; N = 77), and subjects with arterial hypertension (81.4%; N = 364). Details of the clinical assessment of the cohort have been published previously [33].

The genotypes of the rs17366743, rs17300539, rs266729, rs182052, and rs2241766 SNPs of the *ADIPOQ* gene were determined and the results are shown in Table 3. MAFs of the SNPs observed in the study are within the range of MAFs of European populations.

### 3.2. Associations of the ADIPOQ SNPs with Serum Concentrations of Adiponectin

The results of analysis of associations between the SNPs and the circulating adiponectin levels for the most significant genetic models are listed in Table 4.

Regression analysis showed significant association of rs182052 and logarithm of adiponectin concentrations (β = −0.11; 95% CI: −0.19, −0.03; *p* = 0.016). This value is equivalent to a 1.12-fold (1.03–1.21) decrease in the adiponectin levels per copy of the minor A allele. The univariate nonparametric analysis by Mann–Whitney test showed that the rs182052 and rs266729 SNPs are associated with adiponectin concentrations in the dominant and recessive models, respectively. The subjects with the GG genotype (N = 53) of the rs266729 SNP had lower adiponectin concentrations than that in the subjects with the C allele (CC (N = 240) plus CG (N = 154) genotypes). The median adiponectin levels were 6.77 µg/mL (25% 4.97; 75% 9.54) and 7.95 µg/mL (25% 5.80; 75% 12.16) in the subjects with the GG and CC+CG genotypes of rs266729, respectively, and the differences were statistically significant (*p* = 0.04). The subjects with the GG genotype (N = 177) of the rs182052 SNP had higher levels of adiponectin than that in the subjects with the A allele (GA (N = 196) plus AA (N = 74) genotypes). The adiponectin levels were 8.07 µg/mL (25% 6.1; 75% 12.61) and 7.68 µg/mL (25% 5.35; 75% 10.73) in the subjects with the GG and GA+AA genotypes of rs182052, respectively, and the differences were statistically significant (*p* = 0.034). 

No significant associations of rs17300539, rs2241766, and rs17366743 SNPs of the *ADIPOQ* gene with adiponectin concentrations were detected due to insufficient statistical power.

### 3.3. Associations of the rs266729 and rs182052 SNPs with CAD and T2D

Associations of the rs266729 and rs182052 with various diseases were analyzed because these SNPs were associated with adiponectin concentrations. The results are presented in Table 5. Regression analysis of rs182052 showed that the prevalence of unstable angina was significantly increased by 2.55-fold (95%CI 1.4–4.82; *p* = 0.018) and the prevalence of CAD was significantly increased by 1.55-fold (95%CI 1.15–2.09; *p* = 0.021) per copy of the A allele. These associations were confirmed by Fisher’s exact test that detected significantly increased prevalence of unstable angina and CAD in the subjects with the AA genotype of the rs182052 compared with that in the GG and GA genotypes carriers (OR = 3.11; 95%CI 1.20–7.56; *p* = 0.017 and OR = 1.93; 95%CI 1.07–3.64; *p* = 0.024, respectively). 

Prevalence of type T2D was higher in the subjects with the A allele of the rs182052 SNP (OR = 2.29; 95%CI 1.29–4.21, *p* = 0.024) and the association was confirmed by Fisher’s exact tests (OR = 2.1; 95%CI 1.23–3.71; *p* = 0.007) (Table 5).

Regression analysis of rs266729 showed that prevalence of unstable angina was increased by 3.59-fold (95%CI 1.17–10.01; *p* = 0.045) in the subjects with the GG genotype and prevalence of CAD was significantly increased by 1.48-fold (95%CI 1.09–2.03; *p* = 0.045) per copy of the G allele. These associations were confirmed by Fisher’s exact test that detected significantly increased prevalence of unstable angina and CAD in the subjects with the GG genotype of the rs266729 SNP compared with that in the CC and CG genotypes carriers (OR = 3.02; 95%CI 1.04–7.74; *p* = 0.035 and OR = 2.07; 95%CI 1.03–4.48; *p* = 0.033, respectively).

### 3.4. Haplotype Analysis

Two SNPs, rs266729 and rs182052, are highly linked with each other (Dʹ > 0.8; r^2^ = 0.64), while other SNPs are apparently independent (*r*^2^ < 0.05) (Figure 2). Haplotypes were constructed using all five SNPs according to their position in the *ADIPOQ* gene as follows: rs17300539, rs266729, rs182052, rs2241766, and rs17366743. Five haplotypes had frequency >5% and were analyzed as shown in Table 6. The GCGTT haplotype had the highest frequency (48.7%) and was selected as the reference.

The results of regression analysis indicate that the subjects with the second in frequency GGATT haplotype (27.4%) had lower adiponectin levels (β= −0.094; *p* = 0.047) and higher prevalence of unstable angina (OR = 2.054; *p* = 0.041). Moreover, these subjects had higher prevalence of coronary artery disease (OR = 1.546; *p* = 0.014).

Subjects with the GCATT haplotype (9.8%) were characterized by lower adiponectin levels (β= −0.15; *p* = 0.042) and higher prevalence of unstable angina (OR = 3.597; *p* = 0.007).

## 4. Discussion

The results of the present study demonstrate that the rs266729 and rs182052 SNPs of the *ADIPOQ* gene are significantly associated with the circulating levels of adiponectin. Moreover, the minor alleles of these SNPs are associated with the prevalence of coronary artery disease including unstable angina and T2D. These associations may be explained by the antiatherogenic effects of the product of the *ADIPOQ* gene, adiponectin, which regulates vascular endothelial cells and promotes insulin sensitivity [34]. These considerations are indirectly confirmed in the present study which demonstrated significantly lower levels of adiponectin in patients with unstable angina in agreement with the data of meta-analysis [35].

The effects of the rs182052 SNP, which is located in intron 1 of the *ADIPOQ* gene, on the circulating adiponectin levels may be due to a disruption or loss of the Sp1-binding site and a gain of a C/EBP-β-binding site, which may influence the regulation of the expression of the *ADIPOQ* gene [36,37,38]. The effects of the rs266729 SNP may be mediated by changes in the activity of the *ADIPOQ* gene promoter because the minor G allele has been shown to influence DNA-binding activity of the promoter in a 3T3-L1 adipocyte model [39]. Moreover, the minor G allele of the rs266729 SNP may result in a loss of the stimulatory Sp1-binding site [40].

The results of meta-analysis revealed a number of contradictions with regard to associations of the rs266729 SNP and cardiovascular morbidity [7,8]. Apparently, associations between various SNPs and adiponectin concentrations are manifested differentially depending on the ethnic background of recruited cohorts of the subjects. In our study, the rs266729 and rs182052 SNPs were significantly associated with adiponectin levels which are in a perfect agreement with the data [36] obtained in a Chinese population. It should be noted that rs182052 SNP was shown to be associated with adiponectin levels in a European population [4].

Rare, low-frequency SNPs of the *ADIPOQ* gene are poorly studied due to complexity of recruitment of a representative group of subjects with sufficient number of specific minor alleles. For example, the nonsynonymous rs17366743 SNP (MAF < 1.5%) has been shown to be associated with prevalence of T2D and fasting glucose levels, while no significant associations with the circulating adiponectin levels were found [5]. A Framingham Offspring Study [5] demonstrated a significant association of the rs17300539 SNP with adiponectin levels. These SNPs (rs17366743 and rs17300539) were used in the present study and were not associated with adiponectin levels in our cohort. The lack of associations may be due to insufficient power of the present study, which was limited by the number of participants.

We have observed significant linkage between the rs266729 and rs182052 SNPs. Haplotype analysis indicates that the minor allele of the rs266729 SNP is linked with the minor allele of the rs182052 SNP to yield the GGATT haplotype (27.4% frequency). Thus, we were unable to differentiate the effects of these two SNPs directly. However, indirect evidence suggests that certain differences may be possible. For example, the GC**A**TT haplotype (9.8% frequency) includes only the minor allele of the rs182052 SNP and has a more pronounced association with the levels of adiponectin and higher prevalence of unstable angina than those in the subjects with the GGATT haplotype. However, the G**GA**TT haplotype demonstrated a significant association with the prevalence of coronary artery disease, which was higher than that of the GCATT haplotype apparently due to the presence of two minor alleles of the rs266729 and rs182052 SNPs.

## 5. Conclusions

We found that the minor A allele of rs182052 of the *ADIPOQ* gene is significantly associated with a decrease in serum adiponectin levels, and two SNPs (rs182052 and rs266729) of the *ADIPOQ* gene are significantly associated with cardiovascular and metabolic diseases.

## Figures and Tables

**Figure 1 biomolecules-09-00537-f001:**
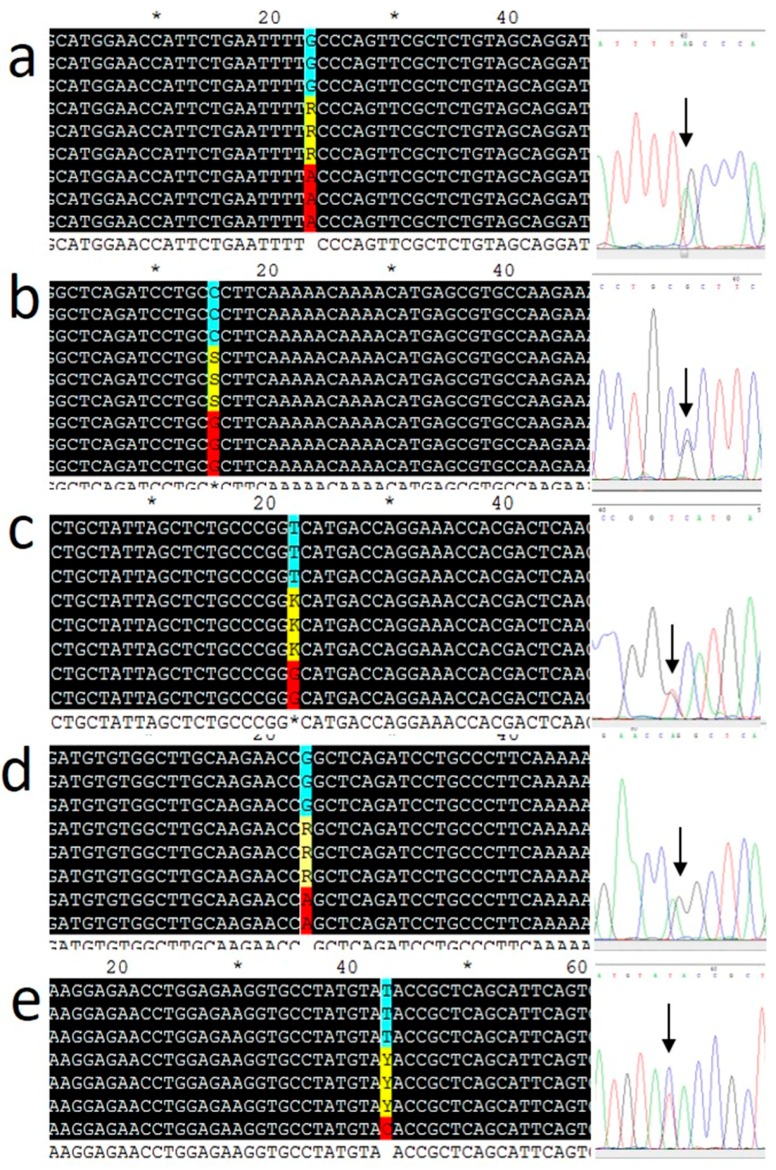
Verification of genotyping results by Sanger sequencing. (**a**) rs182052 (G/A); (**b**) rs266729 (C/G); (**c**) rs2241766 (T/G); (**d**) rs17300539 (G/A); (**e**) rs17366743 (T/C). Arrows on the sequence chromatograms indicate heterozygotes.

**Figure 2 biomolecules-09-00537-f002:**
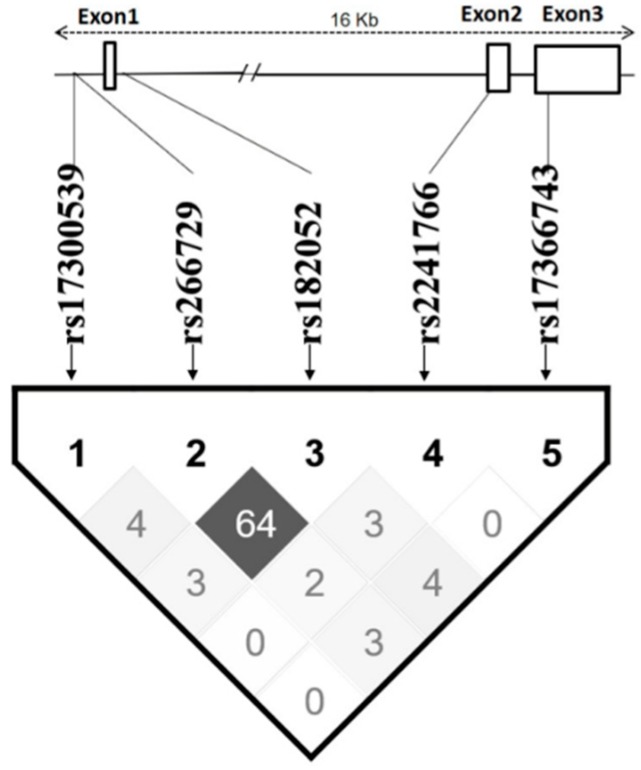
Linkage disequilibrium of the tested SNPs of the *ADIPOQ* gene. The values and colors of the cells correspond to the disequilibrium levels estimated according to the r2, % coefficient.

**Table 1 biomolecules-09-00537-t001:** Primer and probe sequences for the analysis of single nucleotide polymorphisms (SNPs) of the *ADIPOQ* gene.

SNP	Oligonucleotide Primers	Probes	Amplicon Size (bp)
rs17300539	F: 5-TTGAAGTTGGTGCTGGCATC-3	(FAM)-CAGGATCTGAGCCGGTTC-(RTQ1)	193
	R: 5-GGAAGCTGCCACCCACTTA-3	(R6G)-CAAGAACCAGCTCAGATCC-(BHQ2)	
rs266729	F: 5-GTTGGTGCTGGCATC-3	(FAM)-CAGATCCTGCCCTTCAAA-(RTQ1)	127
	R: 5-CCTTGGACTTTCTTGGCACG-3	(R6G)-TGCGCTTCAAAAACAAAACAT-(BHQ2)	
rs182052	F: 5-CCTCCGTTCTCCCAC-3	(FAM)-CCATTCTGAATTTTGCCCAGT-(RTQ1)	145
	R: 5-ACCCTTCCACCTTACTGACC-3	(R6G)-CCATTCTGAATTTTACCCAGTTCG-(BHQ2)	
rs2241766	F: 5-GGATTCCAGGGCTCAGGATG-3	(FAM)-TCTGCCCGGTCATGA-(RTQ1)	139
	R: 5-GCCATCCAACCTGTGCAG-3	(R6G)-TCCTGGTCATGCCCGG-(BHQ2)	
rs17366743	F:5-GGCAGGAAAGGAGAACC-3	(FAM)-AGCGGTATACATAGGCACC-(RTQ1)	180
	R: 5-GTACAGCCCAGGAATGTTGC-3	(R6G)-CTATGTACACCGCTCAGC-(BHQ2)	

F, forward; R, reverse; BHQ2, black hole quencher-2; FAM, 6-carboxyfluorescein; R6G, rhodamine 6G; RTQ1, real-time quencher-1.

**Table 2 biomolecules-09-00537-t002:** General characteristics, biochemical tests, diseases, and medications.

Parameter	Total Group (N = 447)
**general characteristics**	
Age, years	61 ± 9
Weight, kg	85.2 ± 15.0
Body mass index, kg/m^2^	28.73 (26.22–32.7)
Systolic blood pressure, mmHg	131 ± 16
Diastolic blood pressure, mmHg	80 ± 9
Heart rate, bpm	68 (64–74]
**serum biochemical parameters**	
Total cholesterol, mmol/L	4.98 ± 1.27
LDL cholesterol, mmol/L	3.15 ± 1.15
HDL cholesterol, mmol/L	1.00 ± 0.26
Triglycerides, mmol/L	1.56 (1.14–2.12)
Adiponectin, µg/mL	7.87 (5.64–11.81)
Fasting glucose, mmol/L	5.6 (5.2–6.2)
Insulin, µIU/mL	10.5 (7.4–14.5)
HOMA-IR index	2.66 (1.83–3.99)
C-reactive protein, mg/L	2.7 (1.2–5.9)
**diseases**	
Type 2 diabetes mellitus, %	17.2 (13.8–21.1)
Hypertension, %	81.4 (77.5–84.9)
Unstable angina, %	4.9 (3.1–7.4)
Coronary artery disease, %	64.0 (59.3–68.4)
**medications**	
Statins, %	92.8 (90.0–95.1)
Warfarin (anticoagulants), %	7.4 (5.1–10.2)
Clopidogrel (antiplatelet drugs), %	59.7 (55.0–64.3)
Aspirin (antiplatelet drugs), %	89.9 (86.8–92.6)
Angiotensin converting enzyme inhibitors, %	69.6 (65.1–73.8)
Beta-adrenoblockers, %	87.0 (83.6–90.0)
Calcium antagonists, %	27.7 (23.6–32.1)
Diuretics, %	25.3 (21.3–29.6)

Continuous parameters that follow normal distribution are shown as the mean ± SD. Continuous parameters that are not normally distributed are shown as the median [25–75 percentile]. Categorical percentage-based parameters are shown as frequency (95% CI). LDL, low density lipoproteins; HDL, high density lipoproteins; HOMA-IR, homeostatic model assessment of insulin resistance.

**Table 3 biomolecules-09-00537-t003:** Characteristics of the genotyped SNPs of the *ADIPOQ* gene.

SNP	Location on Chromosome 3 ^a^	Relation	Alleles, M/m	Genotypes, N for MM/Mm/mm	Observed MAF, %	MAF	
						Mean MAF, %	Maximal MAF EUR, %
rs17300539	186841671	Promoter	G/A	385/60/2	0.07	0.05	0.07
rs266729	186841685	Promoter	C/G	240/154/53	0.29	0.23	0.32
rs182052	186842993	Intron 1	G/A	177/196/74	0.38	0.39	0.56
rs2241766	186853103	Exon 2 coding synonymous	T/G	391/54/2	0.06	0.05	0.12
rs17366743	186854300	Exon 3 coding nonsynonymous	T/C	432/14/1	0.02	0.02	0.04

**^a^** According to GRCh38.p12 (Genome Reference Consortium human build 38 patch release 12). M, major allele; m, minor allele; MAF, minor allele frequency; MAF EUR, European MAF according to the gnomAD genome database, dataset version 2.1.1.

**Table 4 biomolecules-09-00537-t004:** Univariate nonparametric test and multivariate linear regression analysis of associations of the SNPs of the *ADIPOQ* gene with logarithm of serum concentration of adiponectin.

SNP (MAF, %)	Univariate Wilcoxon–Mann–Whitney Test				Multivariate Linear Regression Parameters				
	Genetic Model	Δ Log (Adiponectin Concentration)	CI 95%	*p*	Genetic Model	β	95% CI	*p*	Adjusted *p*
rs17300539 (7%)	D	0.11	−0.03, 0.25	0.125	D	0.14	−0.03, 0.31	0.103	0.199
rs266729 (29%)	R	−0.16	−0.32, –0.01	0.040 ^a^	A	−0.08	−0.16, 0.01	0.079	0.193
rs182052 (38%)	D	−0.11	−0.22, –0.01	0.034 ^a^	A	−0.11	−0.19, -0.03	0.006 ^a^	0.016 ^a^
rs2241766 (6%)	D	0.11	−0.06, 0.27	0.210	A	0.07	−0.1, 0.23	0.436	0.645
rs17366743 (2%)	R	0.89	−0.92, 3.65	0.151	D	0.05	−0.25, 0.34	0.763	0.728

^a^ Significant associations at *p* < 0.05. A, D, and R, additive, dominant, and recessive genetic models, respectively; Δ, Hodges–Lehman estimator; β, regression coefficient; CI, confidence interval.

**Table 5 biomolecules-09-00537-t005:** Univariate nonparametric test and multivariate logistic regression analysis of associations of the SNPs of the *ADIPOQ* gene with cardiovascular diseases and diabetes.

Diseases		Univariate Fisher Exact Test				Multivariate Logistic Regression Parameters				
	Genetic Model (N for OR)	% in Group with MM/ % in Group with mm	OR	CI 95%	*p*	Genetic Model	OR	95% CI	*p*	Adjusted *p*
	rs266729 SNP									
Unstable angina	R (394/53)	4.1%/11.3%	3.02	1.04–7.74	0.035 ^a^	R	3.59	1.17–10.01	0.018 ^a^	0.045 ^a^
Type 2 diabetes	D (240/207)	14%/21%	1.59	0.97–2.62	0.078	D	1.59	0.93–2.72	0.091	0.091
Coronary artery disease	R (394/53)	67%/79%	2.07	1.03–4.48	0.033 ^a^	A	1.48	1.09–2.03	0.015 ^a^	0.045 ^a^
	rs182052 SNP									
Unstable angina	R (373/74)	3.8%/10.8%	3.11	1.20–7.56	0.017 ^a^	A	2.55	1.4–4.82	0.003 ^a^	0.018 ^a^
Type 2 diabetes	D (177/270)	11%/21%	2.1	1.23–3.71	0.007 ^a^	D	2.29	1.29–4.21	0.006 ^a^	0.024 ^a^
Coronary artery disease	R (373/74)	66%/78%	1.93	1.07–3.64	0.024 ^a^	A	1.55	1.15–2.09	0.004 ^a^	0.021 ^a^

^a^ Significant associations at *p* < 0.05. A, D, and R, additive, dominant, and recessive genetic models, respectively; OR, odds ratio; CI, confidence interval.

**Table 6 biomolecules-09-00537-t006:** Associations of constructed haplotypes with logarithm of serum adiponectin concentration, unstable angina, type 2 diabetes, and coronary artery disease. Haplotypes were constructed using five sequential SNPs of the *ADIPOQ* gene in the following order: rs17300539, rs266729, rs182052, rs2241766, and rs17366743. A total of five haplotypes with frequency >5% were analyzed.

No	Haplotype	Frequency, %	β, Log Adiponectin	OR, Unstable Angina	OR, Type 2 Diabetes	OR, Coronary Artery Disease
1	GCGTT ^a^	48.7	-		-	-
2	GGATT	27.4	−0.094 (*p* = 0.047) ^b^	2.054 (*p* = 0.041) ^b^	1.268 (*p* = 0.266)	1.546 (*p* = 0.014) ^b^
3	GCATT	9.8	−0.15 (*p* = 0.042) ^b^	3.597 (*p* = 0.007) ^b^	1.479 (*p* = 0.247)	1.337 (*p* = 0.287)
4	ACGTT	6.5	0.09 (*p* = 0.301)	0.486 (*p* = 0.496)	1.164 (*p* = 0.711)	1.237 (*p* = 0.508)
5	GCGGT	5.7	0.035 (*p* = 0.706)	0.54 (*p* = 0.565)	1.017 (*p* = 0.968)	0.76 (*p* = 0.314)

^a^ This haplotype was used as the baseline reference. ^b^
*p* < 0.05. β, regression coefficient of multivariate linear regression; OR, odds ratio.

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
