# Peer review of "Associations of SNPs of the ADIPOQ Gene with Serum Adiponectin Levels, Unstable Angina, and Coronary Artery Disease"

_biomolecules, 2019, doi:10.3390/biom9100537_

Round 1

Reviewer 1 Report

Smetnev and colleagues present a study aimed at detecting genetic association between a selected number of SNPs in the ADIPOQ gene and serum adiponectin, CAD and T2D in 447 patients from Russia. The study might be of interest to researchers and physicians specializing in endocrinology, metabolic and cardiovascular disorders. However, in my opinion, the study needs additional data and clarifications.

My comments are listed below in the order of me reading the manuscript.

My first conceptual comment pertains to the nature of the study. From reading the manuscript it appears that this is replication in a separate population rather than a discovery study. To that end, if my summation is accurate, it would have been useful for the authors to have stated that. It is also important for the authors to provide a short and concise description of the previous association studies and provide a more detailed justification, including the authors’ criteria and rationale for selecting these 5 SNPs. My second conceptual comment regards the lack of additional, more in-depth, functional analyses. Such studies will improve the quality of the manuscript as well as provide a better understanding of the potential role ADIPOQ plays in the etiology of levels of serum adiponectin, CAD, and T2D. The authors could have easily perform a gene expression study in their sample to identify whether any of the tested SNPs can be eQTLs that modulates the ADIPOQ expression. If that is not possible, at least they can look at some publically available databases (e.g., GTEx) and see whether there is previous evidence for these SNPs as eQTLs for ADIPOQ. They could have easily performed even a protein QTL (pQTL) analysis using the serum levels of adiponectin and the genotype information for the 5 SNPs. In the statistical section in methods the authors stated that they had validated the regression model using MW t-test. The authors have no provided any justification i) why they need to validate their regression model, and ii) why it is validated using MW test. I don't follow this validation procedure. First, why did the authors feel that they need to validate their parametric regression analysis with a none-parametric t-test? Second, if the authors wanted to validate the regression results, the simplest approach would've been to use a holdout cross-validation method and see how well the statistics hold. Lines 154-158 contain a run-on sentence. I have no idea what the authors are trying to state with it. It is unclear to me the rationale for performing a haplotypic analysis. The authors need to clarify that. Also the authors state that (lines 278-279) that they did not observe an association between adiponectin levels and two of the selected SNPs in their cohort. That could also be a result of lack of power. It would be beneficial if the authors have presented power calculations to demonstrate that a lack of association is not (at least) driven by insufficient statistical power.

Reviewer 2 Report

Authors convincingly present results for association between rs182052 with a decrease in serum adiponectin and prevalence of cardiovascular / metabolic diseases and offer explanations for putative structural changes that can cause such association.

I have just two comments.

Argumentation for a reason why authors did include to the abstract and conclusions, findings of significant association only with rs182052 SNP, though not with rs266729 (though results assume such association to be valid), was not clearly substantiated. Albeit some controversy which exists for rs266729 SNP association with both cardiovascular / metabolic diseases, as was mentioned in Discussion, and reported departure from Hardy-Weinberg equilibrium found in the present work and by others, there are several earlier and recent reports linking rs266729 with both diabetes, i.e. for Arabic and Iranian population (https://www.ncbi.nlm.nih.gov/pubmed/26370001?dopt=Abstract ; https://doi.org/10.1016/j.genrep.2018.11.009; ) and as well for coronary artery diseases, as supported by a most recent meta-analysis (https://doi.org/10.1080/21623945.2019.1595270). What possible explanation can be offered to the observation that other three trial SNPs, that previously were shown to have strong correlations with CAD and T2DM in the various population, notably rs2241766, failed to be reproduced for the selected Russian population?

Round 2

Reviewer 1 Report

I have no additional comments for the authors